# Synthesis of Poly(Ethylene Brassylate-Co-squaric Acid) as Potential Essential Oil Carrier

**DOI:** 10.3390/pharmaceutics13040477

**Published:** 2021-04-01

**Authors:** Aurica P. Chiriac, Alina Gabriela Rusu, Loredana Elena Nita, Ana-Maria Macsim, Nita Tudorachi, Irina Rosca, Iuliana Stoica, Daniel Tampu, Magdalena Aflori, Florica Doroftei

**Affiliations:** 1Department of Natural Polymers, Bioactive and Biocompatible Materials, Petru Poni Institute of Macromolecular Chemistry, 41 A Grigore Ghica Voda Alley, 700487 Iasi, Romania; rusu.alina@icmpp.ro (A.G.R.); lnazare@icmpp.ro (L.E.N.); ntudor@icmpp.ro (N.T.); 2Department of Polycondensation and Thermostable Polymers, Petru Poni Institute of Macromolecular Chemistry, 41 A Grigore Ghica Voda Alley, 700487 Iasi, Romania; macsim.ana@icmpp.ro; 3Center of Advanced Research in Bionanoconjugates and Biopolymers, Petru Poni Institute of Macromolecular Chemistry, 41 A Grigore Ghica Voda Alley, 700487 Iasi, Romania; rosca.irina@icmpp.ro; 4Department of Physical Chemistry of Polymers, Petru Poni Institute of Macromolecular Chemistry, 41 A Grigore Ghica Voda Alley, 700487 Iasi, Romania; stoica_iuliana@icmpp.ro (I.S.); dtimpu@icmpp.ro (D.T.); 5Department of Physics of Polymers and Polymeric Materials, Petru Poni Institute of Macromolecular Chemistry, 41 A Grigore Ghica Voda Alley, 700487 Iasi, Romania; maflori@icmpp.ro (M.A.); florica.doroftei@icmpp.ro (F.D.)

**Keywords:** bio-based copolymers, ethylene brassylate, squaric acid, ring-opening copolymerization, thymol, antimicrobial activity

## Abstract

Bio-based compounds are a leading direction in the context of the increased demand for these materials due to the numerous advantages associated with their use over conventional materials, which hardly degrade in the environment. At the same time, the use of essential oils and their components is generated mainly by finding alternative solutions to antibiotics and synthetic preservatives due to their bioactive characteristics, but also to their synergistic capacity during the manifestation of different biological properties. The present study is devoted to poly(ethylene brassylate-co-squaric acid) (PEBSA), synthesis and its use for thymol encapsulation and antibacterial system formation. The synthesized copolymer, performed through ethylene brassylate macrolactone ring-opening and copolymerization with squaric acid, was physicochemical characterized. Its amphiphilic character allowed the entrapment of thymol (Ty), a natural monoterpenoid phenol found in oil of thyme, a compound with strong antiseptic properties. The copolymer chemical structure was confirmed by spectroscopic analyses. Thermal analysis evidenced a good thermal stability for the copolymer. Additionally, the antimicrobial activity of PEBSA_Ty complex was investigated against eight different reference strains namely: bacterial strains—*Staphylococcus aureus* ATCC25923, *Escherichia coli* ATCC25922, *Enterococcus faecalis* ATCC 29212, *Klebsiella pneumonie* ATCC 10031 and *Salmonella typhimurium* ATCC 14028, yeast strains represented by *Candida albicans* ATCC10231 and *Candida glabrata* ATCC 2001, and the fungal strain *Aspergillus brasiliensis* ATCC9642.

## 1. Introduction

The interest for the current use of essential oils is confirmed by the raising awareness of health and well-being among consumers, and the need to address the issue through compounds of natural sources. The utilisation of the essential oils in various fields has generated the need for their maximum and prolonged beneficial effect, especialy due to their low stability but, also, for their protection from oxygen, light, moisture, and heat. As a result, numerous research studies have been developed to ensure protective covering of essential oils and their complex preparation to successfully manifest their biological properties. Along with natural polymers, the synthetic macromolecular compounds present an alternative to the possibilities of encapsulating essential oils in order to maintain their biological activities, for example, their antioxidant, antibacterial, antiviral, antifungal, anticancer, antidiabetic, and anti-inflammatory properties [1,2,3,4]. A challenge for the use of EOs is represented by the possibilities of their encapsulation with the maintenance of the biological applications in an appropriate polymer structure.

The growing interest for ring-opening (co)polymerization is entirely justified by the potential offered by the bio-based cyclic monomers, for example, cyclic esters, lactides, epoxides, etc., to form degradable or recyclable polymers with a broad range of applications and use especial in biomedical field due to their biodegradability and biocompatibility [5,6,7,8]. The literature in the field has developed very fast with reviews and experimental studies about monomer classes, various groups of cyclic monomers disposed for ROP processes with specific mechanisms including cationic, anionic, enzymatic, coordinative, and radical ring-opening polymerization, or catalysts to achieve ring opening under mild conditions [9,10,11,12,13,14,15]. For example, in Wilson’s review, there is evidence of the advantages brought by the ring-opening polymerization of macrolactones (MLs) derived from sustainable or renewable feed stocks that conduct to new, potentially degradable polymeric materials boasting a diversity of properties and potential applications [7]. Recently studied MLs include ethylene brassylate (EB), a 17-member ring lactone, commercially available and cheaper than other lactides, lactone or macrolactones. The synthesis of EB homopolymer and other various block/copolymers acquired a noticeable interest due to the offered network formation and building blocks capacity, amphiphilic character, self-assembled morphology, rapid crystallization rate, chains easily susceptive to hydrolytic degradation, as well due to the mechanical behavior with improved flexibility and ductility compared to polylactides [16,17,18,19,20,21,22,23,24,25,26,27]. At the same time, ethylene brassylate copolymers have easily found applicability due to the special characteristics conferred by the copolymerization process, which offers an alternative for tuning the properties of the resulting materials [21]. In this context, in the present investigation, ethylene brassylate was copolymerized with squaric acid (SA), a very strong acid suitable for strong intermolecular hydrogen bonds [27]. As a versatile organic molecule, SA was also used in various fields including polycondensation processes or bioconjugates preparation [28,29,30,31,32,33,34].

To our knowledge, up to now, there have been no presented studies on the ring-opening copolymerization procedure of ethylene brassylate with a cyclic comonomer, namely, squaric acid. The novelty of this research is the synthesis of poly(ethylene brassylate-*co*-squaric acid) (PEBSA), as new biodegradable macromolecular compound with improved functionality, and the attempt for encapsulation in its network of the thymol (a natural monoterpenoid phenol found in oil of thyme), a bioactive molecular compound with biocompatible and antioxidant character, strong antiseptic properties, and with antimicrobial activities against *Escherichia coli, Staphylococcus aureus* and *Streptococcus mutans* [35,36,37]. The study presents the synthesis of PEBSA, the physicochemical characterization of the copolymer, and due to the amphiphilic character of the copolymer, preliminary investigation concerning the thymol encapsulation and the antimicrobial activities of the new PEBSA thymol complex in view of its future use as an effective antibacterial system. Thus, not only the coupling of Ty in the polymer matrix is expected, to ensure its protection against degradation in the environmental conditions but, also, due to a good dispersion of the bioactive compound in the polymer network, a good antimicrobial activity for the new synthesized complex.

## 2. Materials and Methods

### 2.1. Materials

All chemicals were used as received without further purification. Ethylene brassylate (EB, 1,4-Dioxacycloheptadecane-5,17-dione, C_15_H_26_O_4_, M_w_ = 270.36 g/mol, purity (Gas Chromatograph > 95.0%), squaric acid (SA, 3,4-dihydroxy-3-cyclobutene-1,2-dione, H_2_C_4_O_4_, M_w_ = 114.06 g/mol, purity (High Performance Liquid Chromatography > 99.0%), and thymol (2-isopropyl-5-methylphenol) were purchased from Sigma-Aldrich Co. (Darmstadt, Germany), anhydrous 1-hexanol was acquired from Across-Organics (Geel, Belgium), dimethyl sulfoxide (DMSO) from Fluka (Buchs, Switzerland), and 1,4 dioxane from Chemical Company Ro. (Iasi, Romania)

### 2.2. Synthesis Procedure

The copolymer synthesis, performed by ring-opening of EB and polycondensing with SA, took place in homogeneous system in 1,4 dioxane, in the presence of 1-hexanol as initiator (EB/alcohol = 10/1 ratio, just related to EB content), and with 75/25 mol/L ratio among EB/SA comonomers. The reaction was conducted in a vessel (where predetermined amounts of the comonomers, initiator and 1,4 dioxane were simultaneously added into the task) immersed in a controlled temperature oil bath at 100 °C, under nitrogen atmosphere, with a stirring rate of 250 rpm, and carried out for 24 h. For example, a specific recipe for the 75/25 EB/SA ratio includes 0.0075 moli = 1.946 mL EB, 0.0025 moli = 0.285 g SA, and 10 mL 1,4 dioxane. After cooling, the reaction mixture was added dropwise into water when the copolymer precipitated, washed several times with water and diethyl ether, and finally freeze–dried and maintained in desiccator at room temperature until further characterization or other processing, as for example the complex PEBSA copolymer thymol (PEBSA_Ty) preparation.

### 2.3. PEBSA Copolymer Characterization

#### 2.3.1. Spectroscopic Analyses

**^1^H-NMR spectra** have been recorded on a Bruker Avance Neo-1 instrument (Bruker BioSpin, Rheinstetten, Germany) operating at 400.1 MHz for ^1^H. The samples were solubilized in DMSO-d6 at room temperature, and transferred in 5 mm Wilmad 507 NMR tubes and recorded with either a 5 mm four nuclei direct detection z-gradient probe for ^1^H. Chemical shifts are reported in δ units (ppm) and were referenced to the internal deuterated solvent DMSO-d6 calibrated at 2.512 ppm, with assignments presented in Figure 1. **^1^H NMR** (400.1 MHz, DMSO-d6, δ, ppm); **EB**: 4.24 (s, 4H, CH_2_-**a**) 2.3 (t, J = 6.87 Hz, 4H, CH_2_-CH_2_-**b**), 1.56 (quintet, J = 6.90 Hz, 4H, CH_2_-**c**), 1.28 (s, 14H, CH_2_-**d**); **SA**: 8.65 (s, 1H, OH). **PEBSA**: 11.61 (bs, 1H, OH), 4.00 (q, J = 5.98 Hz, 4H, CH_2_-**a**), 3.55 (t, J = 5.35 Hz, 2H, -O-**CH_2_**-C=O), 2.19 (t, J = 7.08 Hz, 8.7H, CH_2_-**b**), 1.5 (m, 13.8H, CH_2_-**c**),1.24 (s, 46.8H, CH_2_-**e + d)**, 0.86 (t, J = 7.10 Hz, 3H, CH_3_-**f**).

**FT-IR spectra** of the prepared polymer samples—PEBSA—were recorded on a Vertex Brucker Spectrometer in an absorption mode ranging from 400 to 4000 cm^−1^. The polymer sample was grounded with potassium bromide (KBr) powder and compressed into a disc to analysis. The spectra were acquired at 4 cm^−1^ resolution as an average of 64 scans.

**Raman spectra** of samples were recorded using an inVia™ confocal Raman microscope spectrometer (Renishaw plc, Gloucestershire, UK) equipped with a 633 nm excitation laser line and a Leica DM2700 microscope wih 5×, 20×, 50× and 100× objectives and a Deep Depletion Renishaw CCD Centrus array detector. Scans were accumulated to obtain the spectra in the range of 100 to 3500 cm^−1^ with a resolution of 1 cm^−1^ and a laser power of 17 mW. The background was subtracted and the cosmic ray removed from all spectra.

#### 2.3.2. Thermal Analysis

The synthesized copolymer was investigated from the viewpoint of the thermal behavior. Simultaneous analysis TG/FTIR/MS was used to study the weight losses and identify the main released gases composition. The following system has been used: STA 449 F1 Jupiter thermobalance (Netzsch, Selb, Germany) coupled online with a mass spectrometer Aeolos QMS 403C (Netzsch, Selb, Germany) and FTIR spectrophotometer Vertex 70 (Bruker, Kloten, Switzerland) equipped with TGA-IR external module. The samples weighed up to 10–15 mg, were placed into Al_2_O_3_ crucible and heated under nitrogen flow of 40 mL min^−1^ from room temperature up to 650 °C with a heating rate of 10 °C min^−1^ in the presence of Al_2_O_3_ crucible as reference material. The resulted gases were transferred to mass spectrometer through a quartz capillary of 70 µm maintained at 290 °C. The MS spectra were registered with the aid of Aeolos 7.0 software (Selb, Germany) on the range of *m*/*z* of about 1 up to 200.

Additionally, the gases are transferred through a poly(tetrafluoroethylene) tube maintained at 190 °C, to the TGA-IR external modulus where the FT-IR spectra were recorded on the 600–4000 cm^−1^ interval, at a resolution of 4 cm^−1^ using OPUS 6.5 software.

#### 2.3.3. Size Measurement

The size measurements of samples were performed by a dynamic light scattering (DLS) technique (on a Zetasizer model Nano ZS, Malvern Instruments, Malvern, UK) with a red laser wavelength of 633 nm (He/Ne). The system uses noninvasive back scatter technology (which reduces the multiple scattering effects), wherein the optics are not in contact with the sample, back scattered light being detected. Overall, with the measuring range from 0.6 nm to 6 µm, the system applies the Mie method. DLS measurements yield the Z average of the aggregate’s apparent hydrodynamic diameter (*D_H_*), according to the following equation:(1)$$D_{H}=\frac{\mathrm{kT}}{3\pi \eta D}$$
where *D_H_* is the hydrodynamic diameter, k is the Boltzmann constant, T is the temperature, η is the viscosity, and D is the diffusion coefficient. The hydrodynamic diameter is calculated from signal intensity. DLS study was carried out to investigate the differences registered for PEBSA particle dimension in relation with PEBSA thymol (PEBSA_Ty) bioactive complex.

#### 2.3.4. X-ray Diffraction (XRD) Measurements

X-ray diffraction characterization of the samples was carried out using a D8 ADVANCE (Bruker AXS, Karlsruhe, Germany) device, using the Cu-Kα radiation (λ = 0.1541 nm) and a parallel beam with Göbel mirror. The generator was set at 36 kV and 40 mA, in stepwise mode, (count time 4 s/step, step size 0.02°). The crystallinity index (*C.I.*) was calculated using the L. Segal formula [38]:(2)$$C.I.=\frac{\left(I_{9}-I_{14}\right)}{I_{9}}$$
where *I*_9_ is the maximal intensity of crystal lattice diffraction with 2θ~9°, and *I*_14_ is the minimum intensity of crystal lattice diffraction with 2θ~14°.

#### 2.3.5. Microscopy Analyses

**Scanning electron microscopy** (SEM) studies were performed on PEBSA, and PEBSA_Ty complex samples fixed in advance by means colloidal copper supports. The morphology of the samples was examined with a Quanta 200 Scanning Electron Microscope (FEI) operating at 20 kV in Low Vacuum mode using a secondary electron detector LFD.

**Atomic force microscopy** (AFM) was used for investigation of the film samples resulted from PEBSA, and PEBSA_Ty complex solutions in DMSO. The morphological characteristics were investigated using a Scanning Force Microscope device NTEGRA (NT-MDT, Zelenograd, Moscow, Russia), with high resolution “GOLDEN” Silicon AFM NSG10 Cantilever (resonance frequency ν_res_ = 155 kHz, force constant k = 4 N/m, cantilever length L = 95 ± 5 µm, cantilever width, W = 30 ± 3 µm, cantilever thickness, T = 2 ± 0.5 µm), in atmospheric conditions, at room temperature. Different scan sizes were imaged, starting from 60 × 60 µm^2^ till 1 × 1 µm^2^, with a scanning frequency of 0.3–0.5 Hz. From all investigated areas, the proper scanning dimension was chosen in order to better describe the surface features.

### 2.4. PEBSA_Thymol Bioactive Complex Preparation

The encapsulation of the thymol bioactive compound into the PEBSA polymeric matrix was realized by a co-precipitation technique, realized by an inclusion complexation performed in DMSO by entrapping the thymol into the amphiphilic PEBSA network, followed by the precipitation of the resulted bioactive complex in water. Practically, the complexation went through mixing PEBSA (9 mg/mL DMSO) with thymol (9 mg/mL DMSO) by gentle stirring in a Heidolph rotary evaporator about 2 h (150 rpm and 37 °C). The PEBSA_Ty complex in DMSO was further precipitated in water, and the obtained precipitated complexes were separated by centrifugation and then were freeze–dried. The resulting PEBSA_Ty complex nanoparticles were characterized from the viewpoint of their dimension immediately after preparation, and again after 24 h to conclude about their complexation behavior and stability. The recorded encapsulation efficiency of about 75% was obtained by UV absorption spectrophotometry from the water resulting after washing the PEBSA_Ty complex, by determining the absorbance values recorded at λ = 275 nm, and using the calibration curve previously performed.

### 2.5. PEBSA_Thymol Bioactive Compound Characterization

#### Antimicrobial Activity

The **antimicrobial activity** of the samples (PEBSA 0.008 g/mL, thymol 0.008 g/mL and PEBSA/thymol 1:1) was determined by disk diffusion assay [39] against eight different reference strains: bacterial strains—*Staphylococcus aureus* ATCC25923, *Escherichia coli* ATCC25922, *Enterococcus faecalis* ATCC 29212, *Klebsiella pneumonie* ATCC 10031 and *Salmonella typhimurium* ATCC 14028, yeast strains represented by *Candida albicans* ATCC10231 and *Candida glabrata* ATCC 2001, and the fungal strain *Aspergillus brasiliensis* ATCC9642. All microorganisms were stored at −80 °C in 20–40% glycerol. The bacterial strains were refreshed in tryptic soy agar (TSA) at 36 ± 1 °C. The yeast strains were refreshed in YM agar and the fungal strains was refreshed on potato dextrose agar (PDA) at 25 ± 1 °C. Microbial suspensions were prepared with these cultures in sterile solution to obtain turbidity optically comparable to that of 0.5 McFarland standards. Volumes of 0.2 mL from each inoculum were spread on the Petri dishes. The sterilized paper disks (6 mm) were placed on the plates and an aliquot (20 µL) of the samples was added. To evaluate the antimicrobial properties, the growth inhibition was measured under standard conditions after 24 h of incubation at 36 ± 1 °C for the bacterial and the yeast strains and after 48 h at 25 ± 1 °C for the fungal strain. All tests were carried out in triplicate to verify the results. After incubation, the diameters of inhibition zones were measured by using Image J software [40].

The minimum inhibitory concentration (MIC) was determined by broth dilution method performed in 96-well microtiter plates. Bacterial culture grown to log phase was adjusted to 1 × 10^8^ cells/mL in Mueller-Hinton (MH) Broth, RPMI and Potato Dextrose Broth. Inoculants of 50 µL were mixed with 50 µL of serial dilutions of samples and were subsequently incubated at 36 ± 1 °C for 24 h. The antibacterial activity was determined on the basis of turbidity by a FLUOstar^®^ Omega microplate reader (BMG LABTECH, Ortenberg, Germany). The minimum bactericidal concentration (MBC) was determined by spreading of 10 µL samples from wells on MH agar plates. The concentration of sample that prevented the growth of bacteria was recognized as MIC, and the lowest concentration that destroyed all the microbial cells was considered as MBC. In order to determine MIC/MBC values, the experiments were performed in triplicate. The determination of minimum inhibitory concentration for *A. brasiliensis* was done after 48 h of incubation by removing 10 µL of the contents from wells showing no visible growth and spreading them on to Potato Dextrose agar plates. The plates were then incubated for 72 h and MIC was determined as the lowest drug concentrations which killed 95% of the inoculum.

### 2.6. Statistical Analysis

All data were expressed as the mean ± standard deviation of the mean. Statistical analysis was performed XLSTAT software [41].

## 3. Results and Discussion

The increased demand for biobased materials including synthetic macromolecular compounds is due to the numerous advantages associated with their use over conventional materials, which hardly degrade in the environment, and thus threaten the atmosphere. The PEBSA preparation (synthesis and self-assembly schematized in Figure 2) is part of this direction, and even we do not take into account the other special features the copolymer is offering, it is worth studying this compound owing to its capacity for complexation and structuring.

### 3.1. ^1^H NMR Spectra

The ^1^H-NMR spectrum of PEBSA illustrated in Figure 3, certifies the copolymer synthesis.

### 3.2. FTIR Spectra

Figure 4 introduces the FTIR spectra of the investigated compounds from this study; namely, PEBSA copolymer, PEBSA thymol complex (PEBSA_Ty), and thymol, while in Table 1 are presented the spectral assignments for the registered frequencies. FTIR spectrum of PEBSA also confirms the synthesis of the copolymer through the registered modified peaks compared to the comonomers spectra, especially in the 2850–3000 cm^−1^ and 1600–1700 cm^−1^ domains, and attributed to the polymerization process and bonds between the macromolecular chains.

PEBSA spectrum present peaks corresponding to EB and SA comonomers as well as the changes occurring as a result of the opening of the macrolactone cycle, which are also registered. Thus, the specific peaks of OH from the squaric acid are no longer found in the copolymer spectrum, which attests to the formation of PEBSA. Additionally, asymmetric and symmetric CH_2_ bonds and carbonyl bonds (C=O), are found at 2926.33, 2855.14 and 1698.44 cm^−1^, respectively. Asymmetric and symmetric CH_2_ bonds and carbonyl bonds (C=O), are found at 2926.33, 2855.14 and 1698.44 cm^−1^, respectively; C-O and C-C bands at 1240.99 and 1231.46 cm^−1^, while asymmetric symmetric C-O-C bands are present at 1183.31 cm^−1^. Additionally, C=C bending from squaric acid are found at 919.18, 681.97, and 531.26 cm^−1^, respectively.

### 3.3. Raman Spectra

Raman spectroscopy is one of the most important techniques for the analysis of various properties of polymeric systems obtained from their vibrational properties, and becomes, in this context, a complementary method of infrared absorption. The Raman spectra, which also provides information on the functional groups present in the macromolecular chains, is more proper for examining chain conformation considering usually intense backbone vibrations. Figure 5a,b and Figure 6 illustrate the Raman spectra of squaric acid and of the PEBSA copolymer.

As can be observed from the Raman spectra of PEBSA, the registered bands at the following wavenumbers 236, 304, 379, 633 (torsional motions, which are out-of-plane bending about the C=O and C-C), 725 (H-bond), 1048 (ν(C-C)), 1171 ((ν(C-C), (ν(C=C)), 1299 ((ν(C=O)), 1510 ((ν(C=O)), 1613 ((ν(C=O)), and 1824 cm^−1^ ((ν(C=O)), are in agreement with SA vibrational assignments from the literature, and thus the copolymer chemical structure is confirmed [42,43]. The emerged new characteristic bands from 2843, 2858, 2878, 2904, 2935 cm^−1^ in PEBSA Raman spectra are attributed to the intra- and intermolecular physical interactions within the macromolecular chains. Additionally, we must note the absence of peaks in the 1300–1800 cm^−1^ range, which are representative and expected for OH stretching inside SA, from the Raman spectra of the copolymer. This absence confirms the copolymerization of EB with SA as it is illustrated in the schematic representation of PEBSA chemical structure (Figure 2).

Figure 7 illustrates the PEBSA polymer surface structure. A homogeneous surface can be observed, and this image allows distinguishing among the discrete presence of the comonomers into the polymer matrix. Thus, the polymeric network of PEBSA is characterized by a relatively ordered distribution, with a good spatial position and arrangement between EB and SA comonomers.

All the performed spectroscopic analyses confirm the synthesis of PEBSA copolymer, from the breaking cycle of EB evidenced by NMR spectra, to correlated and complementary peaks from FTIR and Raman spectra, which attest the asymmetric and symmetric CH_2_ and carbonyl bonds presence from chains, as well the intra- and intermolecular physical interactions within the macromolecular chains. These results are also sustained by other investigations [23].

### 3.4. PEBSA Thermal Behavior

The TG and DTG curves that resulted after thermal characterization of PEBSA are presented in Figure 8. As can be seen, the thermal degradation of the copolymer takes place in 3 stages with significant mass losses due to the thermal degradation of EB and SA units from the copolymer, respectively 26.67% with a maximum degradation at T_peak1_ = 288 °C in the first stage, 45.86% (T_peak2_ = 348 °C) in the second stage, and 25.15% (T_peak3_ = 441 °C) in the last stage. Up to 650 °C PEBSA was almost entirely degraded by about 98%. Analyzing the thermal stability of PEBSA, compared to EB and SA it was found that the copolymer has an intermediate stability located between the two components. Comparative data concerning the thermal behavior of poly(ethylene brassylate) homopolymer, mentioned by literature in the field, presented the initial temperature for weight loss of about 250 °C in case of the homopolymer instead of 288 °C for PEBSA, while at 650 °C the copolymer is degraded in proportion of 98% compared to homopolymer which reached zero at around 480 °C [16]. These temperatures increases are due to the appearance of physical bonds between macromolecular chains, which can generate supramolecular polymeric structure. In this sense, additional investigations are currently being carried out.

The gases evolved during thermal degradation were recorded by FTIR and MS analyses and identified using IR spectra and MS signals available in NIST libraries [44]. As it can be seen in 3D spectrum (Figure 9a) the maximum intensity of the absorption bands of the gasses appears at temperatures close to the DTG peaks, and is the result of the pyrolysis processes, chain scission and the fragmentation or recombination reactions that take place at high temperatures. FTIR spectra of the gasses evolved at the maximum temperature according to Gram Schmidt plot (290, 365 and 460 °C) are presented in (Figure 9b), and the main absorption bands appear in the following regions: 3851–3575 cm^−1^, 2928–2861 cm^−1^, 2356 cm^−1^, 2179–2108 cm^−1^, 1714, 1645 cm^−1^, 1532–1455 cm^−1^, 1380 cm^−1^, 1160 cm^−1^, and 733–672 cm^−1^. The wide absorption band recorded at about 3250 cm^−1^ is attributed to the MCT detector (ice band) of the TGA-IR external module cooled with liquid nitrogen [45].

The absorption bands located between 3851–3575 cm^−1^ and 1380–1160 cm^−1^ can be assigned to water vapors and alcohols that can appear at the thermal degradation of the secondary hydroxyl or ester groups. The signals located at 2928, 2861, 1645, 1532, 1455 cm^−1^ are assigned to the vibration of CH, CH_2_ and CH_3_ groups located in the chemical structure of the alkanes, cycloalkanes and alkenes. The strongest signal present at 2356 cm^−1^ is attributed to carbon dioxide, which can occur as result of the ester group degradation and one small signal at 2179–2108 cm^−1^ assigned to carbon monoxide [46]. The signal located at 1714 cm^−1^ (νCO groups) and 2860 cm^−1^ (νCH groups) can be assigned to the aldehydes and acids products occurring at thermal degradation of the copolymer to high temperatures. These data are in agreement with the chemical structure of PEBSA. The data obtained from FTIR analysis were also confirmed by MS data presented in Figure 10. Thus, the ionic fragments located up to 120 amu, at temperatures of 284 and 458 °C are assigned as it follows: water (*m*/*z* = 18, 17, 16), carbon dioxide (*m*/*z* = 44, 28, 22, 16, 12), carbon monoxide (*m*/*z* = 28, 16, 12), acetaldehyde (*m*/*z* =44, 43, 42, 29, 15, 14), butene (*m*/*z* = 56, 41, 39, 29, 28, 27), cyclopropane (*m*/*z* = 42, 41, 40, 3938, 27, 26), cyclohexane (*m*/*z* = 84, 69, 56, 55, 41, 39, 27), ethanol (*m*/*z* = 46, 45, 43, 31, 29, 27, 17, 15).

### 3.5. DLS Measurements

The interest in the dimensional evaluation of the synthesized copolymer derives mainly from its use as a polymeric network for the incorporation of a bioactive compound, a process that takes place preferably in solution. As illustrated in Figure 2, the synthesized PEBSA copolymer has the ability to self-assembly through physical bonds. This fact is confirmed by the dimensional analysis performed by DLS. Figure 11 illustrates the size of PEBSA, PEBSA_Ty complex and thymol registered by DLS analysis. Clear differences between the sizes of the investigated compounds are put into evidence. Thus, the hydrodynamic size measurement of PEBSA particles shows two populations: one at 420 nm and another at around 5000 nm.

The increase in the PEBSA size distribution up to 5000 nm is considered the result of self-assembly of the copolymer macromolecular chains through strong attractive van der Waals interactions.

### 3.6. X Ray

The X-ray diffractograms of PEBSA and SA are presented in Figure 12. The SA diffractogram presented in this figure illustrates three characteristic bands at ca. 22.0°, 34.0° and 37.0° for this comonomer, similar peaks being reported by other authors too [47,48,49]. Instead, the powder XRD pattern of PEBSA copolymer exhibits an increased number of peaks at 2θ values, respectively, at 5.5°, 12.2°, 18.3°, 22.2°, 22.8°, 26.2°, 27.9°, 33.6°, 36.8°, but even if the number of peaks is increased they present a diminished intensity. It should also be noted that while for a degree of crystallinity of about 89.9% found for SA with the crystallite size of 24.39 nm, PEBSA shows a degree of crystallinity of about 70.5% while the crystallite size is of 20.76. SA’s ability to crystallize has been presented in the literature, when the possibility of the formation of polyanionic supramolecular architectures suited to interact with molecules rich in hydrogen-bond donors was also mentioned [50].

These data again confirm the chemical structure of the synthesized copolymer as well as the existence of physical bonds between the PEBSA macromolecular chains that generate self-assembling behavior. Moreover, the peak at 5.5° and its size could be the confirmation of a supramolecular structure for PEBSA, but in this sense further investigations are needed.

### 3.7. Microscopy Analyses

#### Scanning Electron Microscopy Studies

The microstructure and morphologies of the PEBSA polymeric matrix and PEBSA_Ty complex were evaluated by scanning electronic microscopy (SEM). Figure 13 shows the surface of the investigated samples.

As can be observed, the PEBSA microstructure illustrates smoother surface with a few fine holes and relatively uniform distribution of tangled lines, aspect which is attributed to the presence of the attractive intermolecular interactions between the PEBSA polymeric chains, which lead to a network structure physical crosslinked.

### 3.8. AFM Studies

Amphiphilic character of a polymer structure constitutes a key factor for the subsequent ability of the network to encapsulate different active compounds depending on the required applicability. The surface roughness of the polymer can recommend or not further entrapping capacity, and atomic force microscopy allows the investigation of the topographic characteristics of the surfaces. The surface morphology characteristics of PEBSA copolymer investigated by AFM are illustrates in Figure 14a. The 2D height image of the investigated sample revealed topography concretized in a relief with hills and valleys determining a surface root mean square roughness (Sq) of 240 nm capable for hosting new molecules guest, for example. The core fluid retention index (Sci) and core void volume (Vvc), calculated based on the bearing analysis using Abbott–Firestone curves, were 1.590 and 0.305 μm, respectively. Entities with regular structure, are uniformly distributed all over the surface, with variable dimensions (individual of about 400–500 nm, but also crowded, their dimensions varying up to a few micrometers), and which are attributed to the presence of squaric acid in the composition of the poly(ethylene brassylate-co-squaric acid). Similar values can be observed in DLS measurements. The distribution of the two components of the studied copolymer was also highlighted by means of phase-contrast AFM images (as the one from Figure 14b), acquired simultaneously with the topographic data, allowing surface structure and material domains to be directly compared. This chemical mapping of the surface was sensitive to the differences in surface stiffness/softness of the material.

### 3.9. PEBSA_Thymol Bioactive Complex

Thymol (2-isopropyl-5-methylphenol) is one of the main compounds of thyme essential oil with large applicability in biomedical field due to its anti-inflammatory, antiviral, antibacterial, and antiseptic properties [35,36,37]. Additionally, recent investigations have demonstrated their antifungal, antiviral, antibiofilm, and anticancer properties [51,52]. The encapsulation of Ty in PEBSA network is illustrated in Figure 15.

Van der Waals physical bonds ensure the coupling of the bioactive compound in the polymeric matrix, and the physicochemical characterization confirms the formation of the bioactive complex. Thus, the supplementary peaks appeared in the FTIR spectra of PEBSA_Ty (Figure 4 and Table 1) positioned at 3397.74 (O-H stretching intermolecular bonds), 1623.73 (C=C stretching), 1520.41 (C=C stretching), and 1087.00 (C-O stretching) cm^−1^, are attributed to the thymol presence and attest the formation of the bioactive complex.

The size measurement of PEBSA_Ty particles (Figure 11) evidences the affinity between the copolymer and the bioactive compound, concretized in the diminution of the hydrodynamic diameter of the complex 24 h after preparation. If the value of the hydrodynamic diameter of PEBSA can be up to 5000 nm and of Ty of about 550.6 nm, in the case of the PEBSA_Ty complex, it presents only 385.1 nm. Thus, the hydrodynamic diameter of PEBSA_Ty decreases after structuration of the complex through cumulative formation of the physical bonds, namely, H bonds between the OH group of thymol and the C=O groups from PEBSA, and other van der Waals bonds.

The microstructure of the PEBSA_Ty complex evaluated by SEM evidences a homogenous aspect with well distributed Ty particles embedded in the PEBSA polymeric matrix Figure 13b,c. Thus, the morphological changes in the PEBSA structure consist in the uniform distribution of Ty in the polymer matrix (illustrated as well in schematic representation from Figure 15), which can be seen as white spherical particles. The homogeneous distribution of Ty in the PEBSA network is due to the amphiphilic character of the copolymer, and also, to the chemical attraction between the macromolecular chains and the bioactive compound capable of establishing physical bonds.

The surface morphology of the PEBSA-Ty complex Figure 14e,f as well as of Ty Figure 14c,d was also highlighted by the AFM measurements. Thus, the 2D height and phase images from Figure 14c,d revealed the presence of almost spherical and uniform in shape thymol particles, with a mean diameter of 640 ± 60 nm, evaluated using multiple height profile measurements. These values are also comparable with those obtained from the DLS measurements. The small contrast in the phase image in Figure 14d, induced only by the height changes, indicates that the sample is homogenous, composed of a single component, inducing a small root mean square roughness of 13 nm, comparing to the copolymer matrix.

The topographical images corresponding to PEBSA-Ty complex (Figure 14e) confirm the observations resulted from SEM micrographs. The surface root mean square roughness of 184 nm, calculated over the 30 × 30 μm^2^, was smaller than that obtained over the same area for the PEBSA polymer matrix, due to the encapsulation phenomenon, which also reduces the core fluid retention index and core void volume to 1.491 and 0.227 µm, respectively. The high phase contrast images have revealed the arrangement of the Thymol in the PEBSA network, the phase shifts being registered as bright regions (corresponding to the PEBSA) and dark regions (corresponding to the Ty).

### 3.10. Antimicrobial Activity of PEBSA_Ty Complex

The antimicrobial activity of Ty is known, and the bactericidal mechanism of the membrane disruption has also been identified [53]. Additionally, its use does not increase a potential induction of bactericidal resistance, presenting lower cell toxicity than the conventional widely used chlorhexidine. More than that, significant biostatic effects were observed at sub-MIC levels [54].

The antimicrobial activities of the new prepared PEBSA_Ty complex are presented. Thus, the data on the diameters of the inhibition zones (mm) are presented in Table 2 and Figure 16a–h. PEBSA polymeric matrix did not present antimicrobial activity against none of the reference strains. Ty used presented a very small antimicrobial activity against *S. aureus*, *E. coli*, *E. faecalis*, *S. Typhimurium, C. albicans* (up to 10 mm of inhibition zone)*,* but was efficient against *K. pneumoniae* and *A. brasiliensis* (up to 18 mm of inhibition zone). PEBSA_Ty proved to be slightly more efficient than Ty alone against the bacterial strains represented by *S. aureus*, *E. coli*, *E. faecalis*, *K. pneumoniae*, and against the fungal strain represented by *A. brasiliensis* (as presented in Table 2).

MIC and MBC of PEBSA samples were not determined due to the fact that the compound did not present antimicrobial activity. MIC for all the other tested samples was of 0.008 g/mL against all the tested microorganisms, excepting *C. glabrata*. MBC could not be determined for none of the tested combinations (Table 3).

As PEBSA_Ty proved to be slightly more efficient than Thymol alone against the bacterial strains represented by *S. aureus*, *E. coli*, *E. faecalis*, *K. pneumoniae*, and against the fungal strain represented by *A. brasiliensis* (as presented in Table 2) it can be concluded on the advantages conferred by Ty coupling in the PEBSA polymeric matrix, which confers protection to the bioactive compound concretized in a gradual release thereof, and thus ensures a prolongation of the antimicrobial effect materialized in an increase in the antimicrobial activity.

## 4. Conclusions

The study presents, for the first time, the synthesis of poly(ethylene brassylate-co-squaric acid), a biobased product prepared by polycondensing of ethylene brassylate with squaric acid, and also, the utilisation of the copolymer for encapsulation of thymol known for its antioxidant character, strong antiseptic properties, and antimicrobial activities. The performed spectroscopic analyses, ^1^H-NMR, FTIR, and Raman, confirmed the chemical structure of the copolymer. Additionally, the thermal analysis evidenced good thermal stability for the copolymer with a maximum degradation at T_peak1_ = 288 °C. The DLS measurements show two populations of particles with dimensions of about 420 nm and another at around 5000 nm, which were justified by the ability of PEBSA to self-assembly through van der Waals bonds that can lead to a supramolecular structure fact evidenced by X-ray measurements as well, but further investigations in this context are in course. The microscopy analyses illustrated the homogeneous aspect of the copolymer network, and as well of the complex realized between PEBSA and thymol. The encapsulation capacity of PEBSA was confirmed by coupling Ty. The PEBSA_Ty complex formation was confirmed by FTIR spectra, SEM and AFM images, and DLS measurements. The antimicrobial activity of the new complex was investigated against eight different reference strains. Clear advantages are brought by using PEBSA network for coupling Ty which are concretized in the increase in the antimicrobial activity with around 10%, especially when PEBSA_Ty complex was used against *Staphylococcus aureus ATCC25923* (7.7%)*, Escherichia coli ATCC25922* (~5%)*, Enterococcus faecalis ATCC 29212* (34%)*, Klebsiella pneumonie ATCC 10031* (7%)*,* and *Aspergillus brasiliensis ATCC9642* (11%). The biobased nature of the copolymer and the antimicrobial character conferred to PEBSA_Ty after thymol encapsulation recommends the new complex for supplementary investigations for its potential use as an effective antibacterial system.

## Figures and Tables

**Figure 1 pharmaceutics-13-00477-f001:**
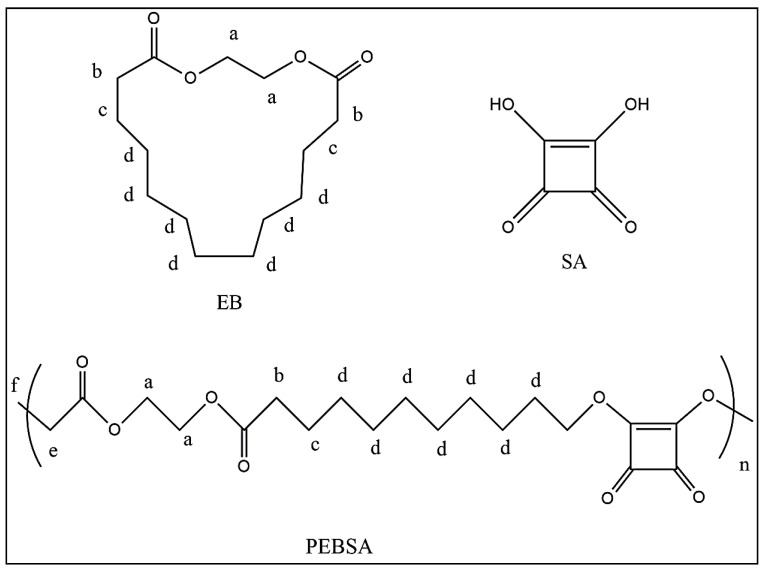
EB, SA and PEBSA assignments in ^1^H-NMR spectra.

**Figure 2 pharmaceutics-13-00477-f002:**
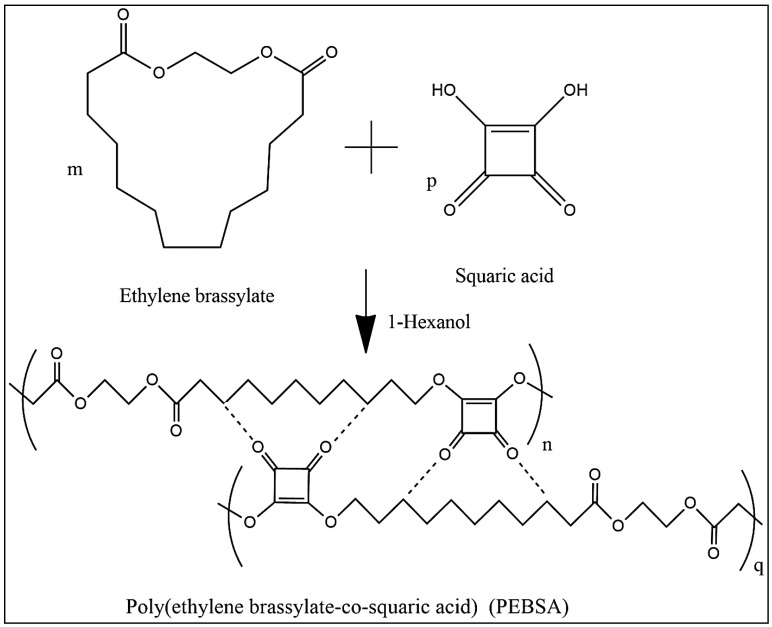
Schematized illustration for the synthesis and self-assembly of the PEBSA chains.

**Figure 3 pharmaceutics-13-00477-f003:**
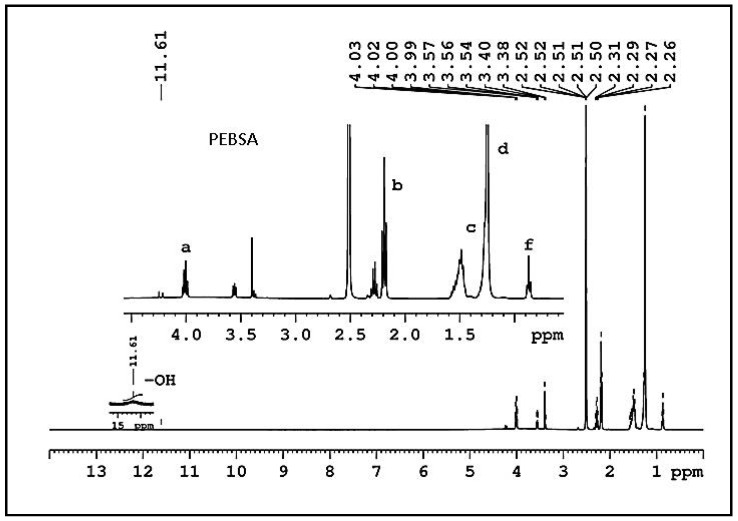
PEBSA ^1^H-NMR spectrum. Thus, the appearance in spectra of the signal from 0.86 ppm attribute of methylene protons confirms the dissolution/breaking of the EB cycle and the formation of PEBSA copolymer through copolymerization with SA comonomer. The repeating units of EB are also confirmed by the presence of chemical shifts at 2.19 ppm (peak (**b**)), 1.50 ppm (peak (**c**)), 1.24 ppm (peak (**d**)), and 4.00 ppm (peak (**a**)). The signal at 4.00 ppm (peak (**a**)) is ascribed to the methylene protons of EB main chain, whereas signals at 11.6 ppm (-OH, from SA) and at 0.86 ppm (peak (**f**)) of methylene protons end group.

**Figure 4 pharmaceutics-13-00477-f004:**
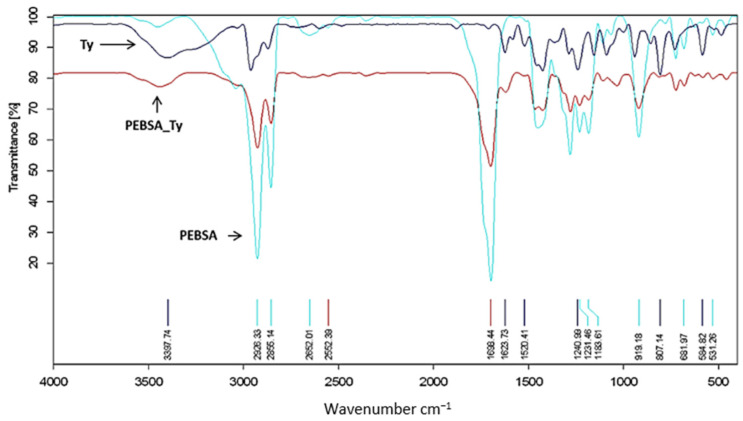
PEBSA, PEBSA_Ty, and thymol FTIR spectra.

**Figure 5 pharmaceutics-13-00477-f005:**
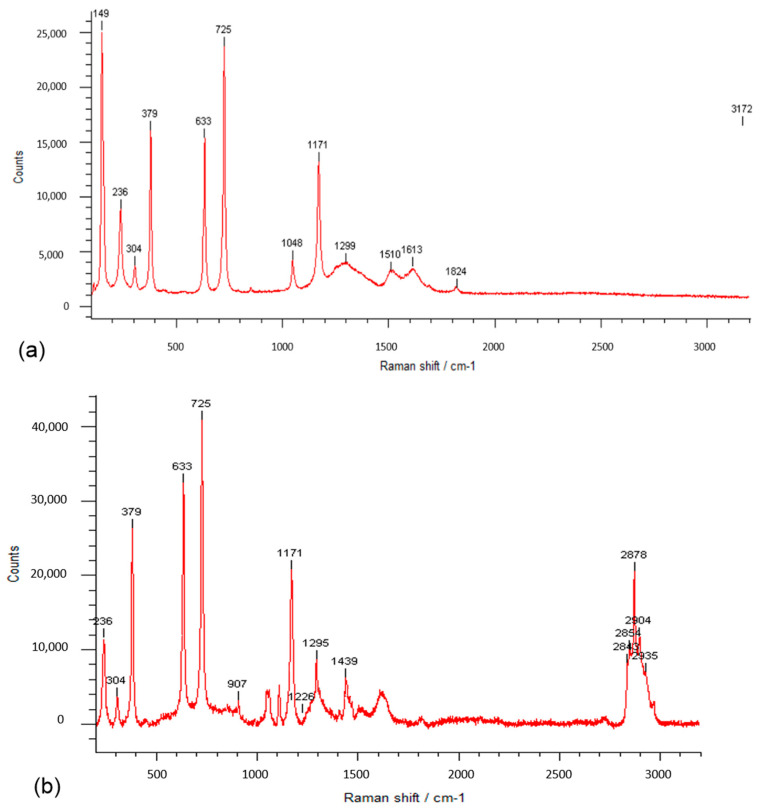
Raman spectra of squaric acid (**a**) and PEBSA copolymer (**b**).

**Figure 6 pharmaceutics-13-00477-f006:**
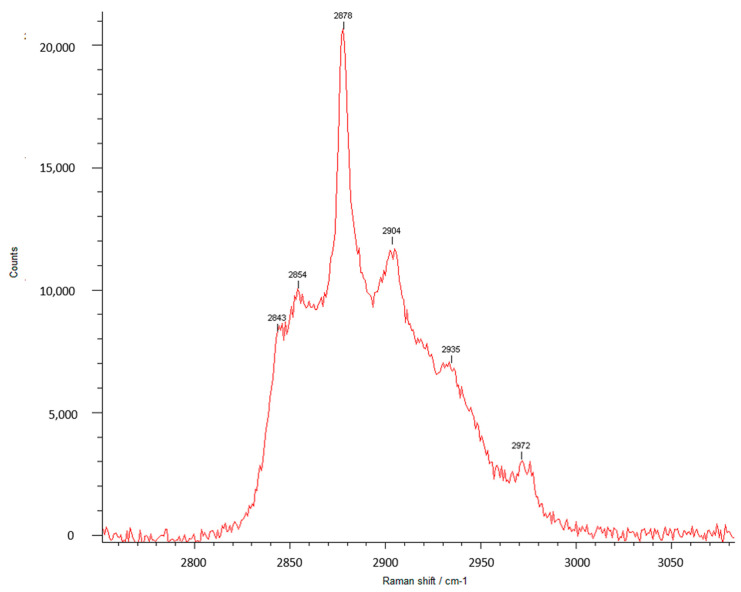
Raman spectra of PEBSA in 2800–3000 cm^−1^ narrow spectral region.

**Figure 7 pharmaceutics-13-00477-f007:**
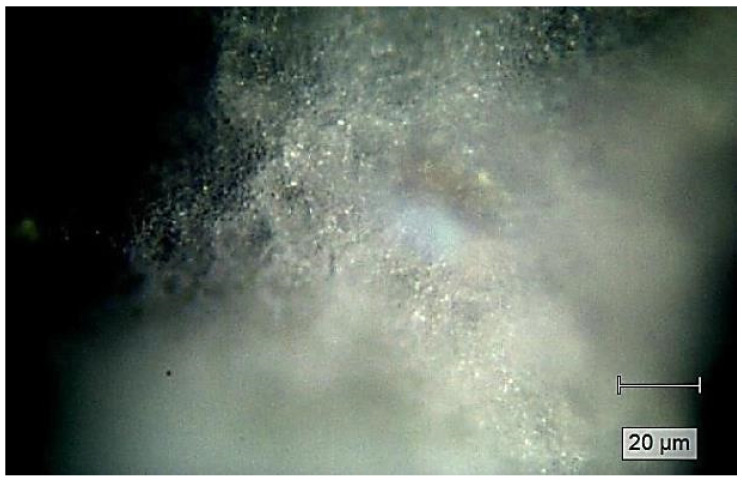
Raman imaging of PEBSA (75/25).

**Figure 8 pharmaceutics-13-00477-f008:**
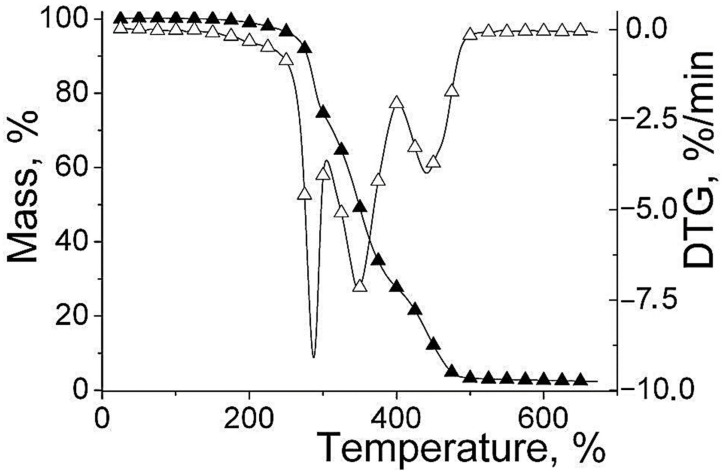
TG and DTG curves of PEBSA.

**Figure 9 pharmaceutics-13-00477-f009:**
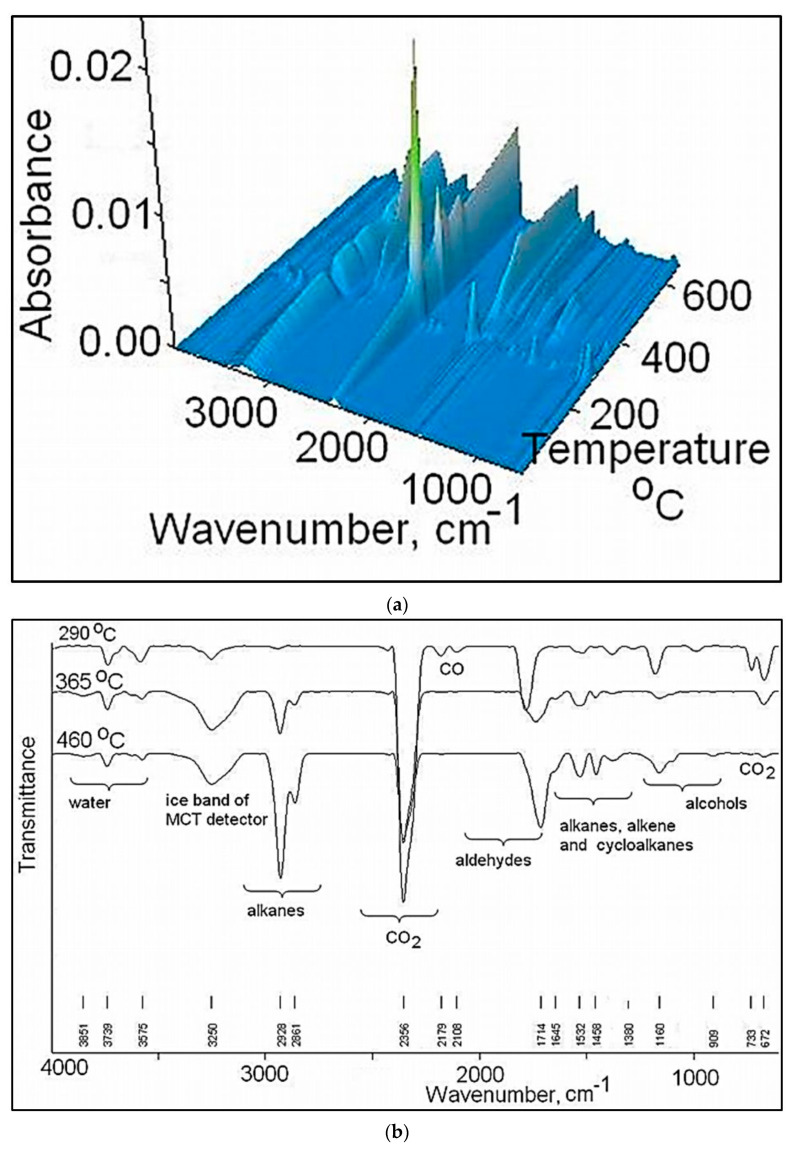
Stacked plot diagram (**a**) and FTIR spectra of the evolved gases at 290, 365, and 460 °C (**b**) for EBSA copolymer.

**Figure 10 pharmaceutics-13-00477-f010:**
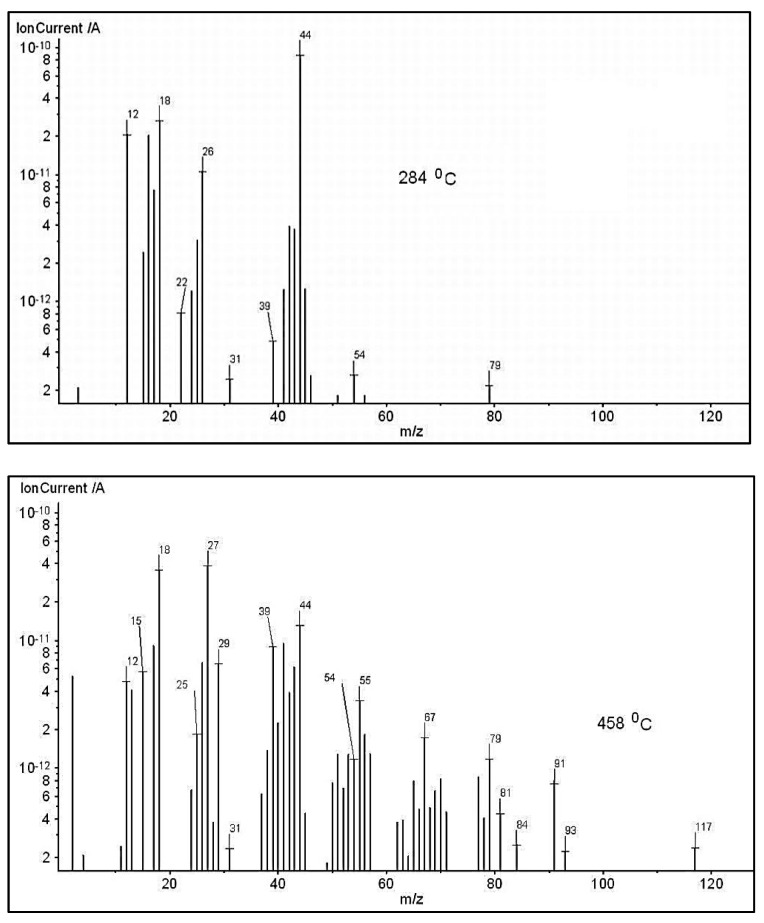
MS spectra of thermal degradation products recorded at 10 °C min^−1^ (at the maximum temperature of the evolved gases) for PEBSA copolymer.

**Figure 11 pharmaceutics-13-00477-f011:**
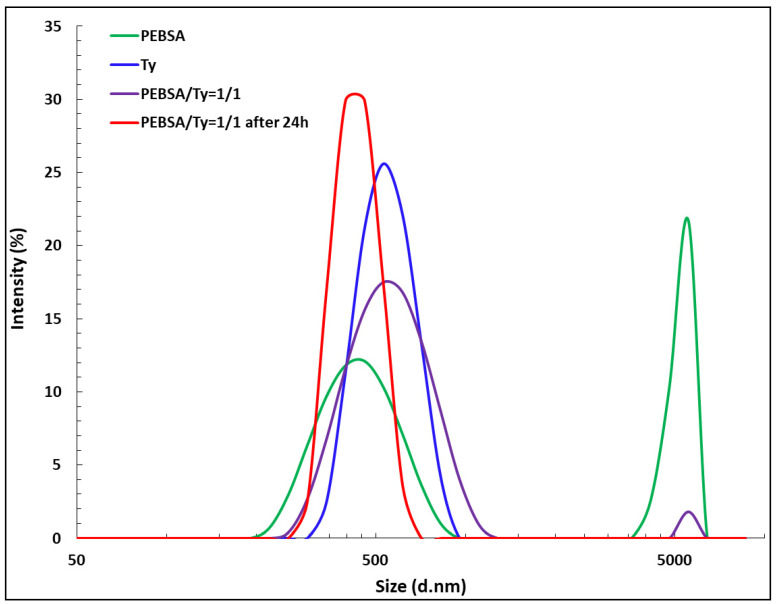
Size distribution of PEBSA copolymer, PEBSA_Ty complex and Thymol.

**Figure 12 pharmaceutics-13-00477-f012:**
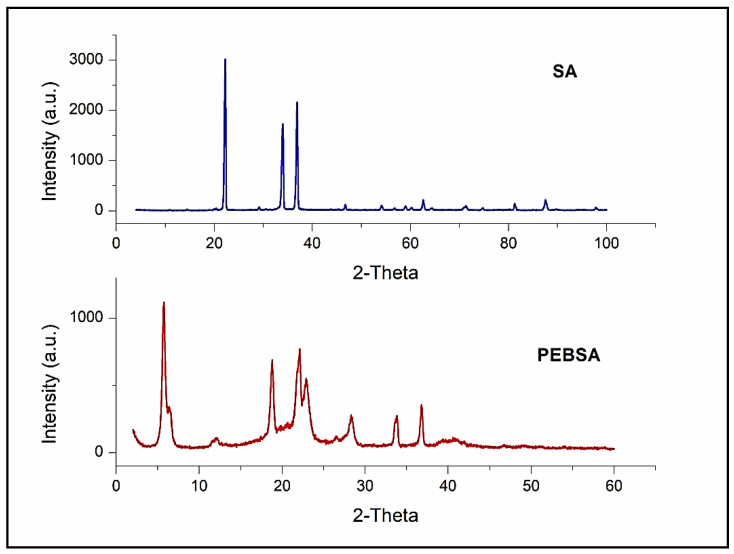
XRD of PEBSA and SA comonomer.

**Figure 13 pharmaceutics-13-00477-f013:**
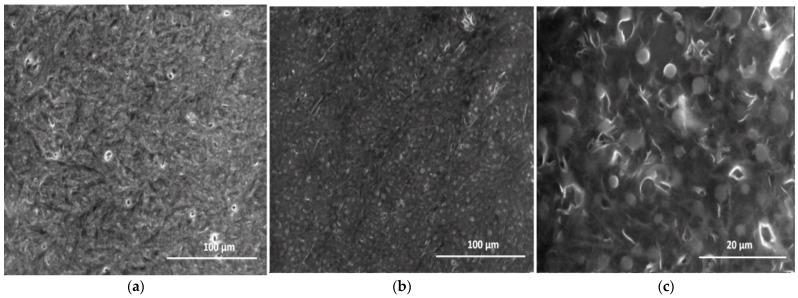
SEM micrograph for the microstructure of the PEBSA polymer matrix (**a**) and of PEBSA_Ty complex (**b**,**c**).

**Figure 14 pharmaceutics-13-00477-f014:**
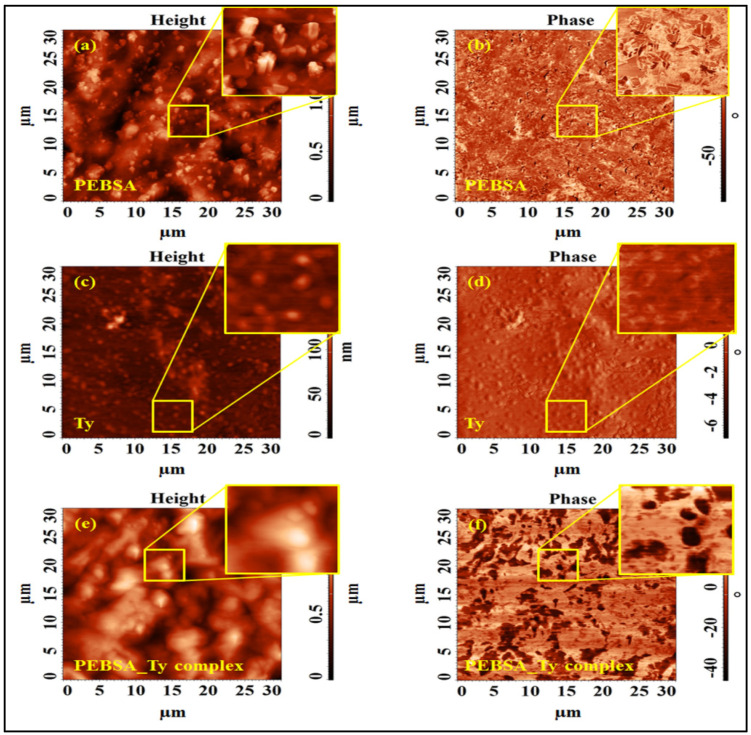
Bi-dimensional and phase contrast AFM images collected on PEBSA copolymer (**a**,**b**), thymol (**c**,**d**) and PEBSA-Ty complex (**e**,**f**).

**Figure 15 pharmaceutics-13-00477-f015:**
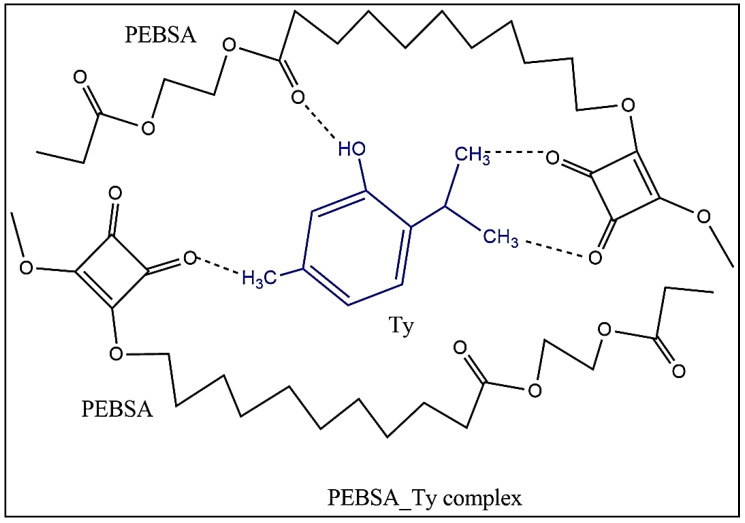
Illustration of PEBSA_Ty bioactive complex formation.

**Figure 16 pharmaceutics-13-00477-f016:**
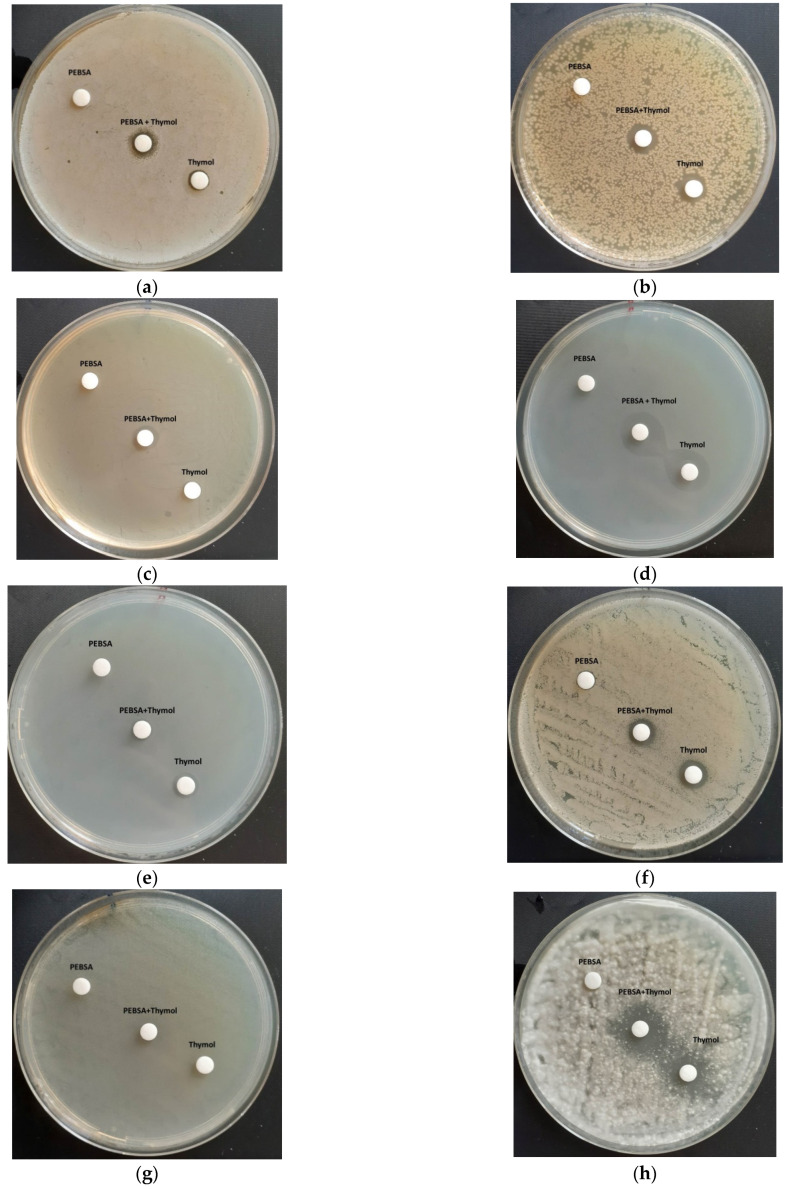
Antibacterial activity of the tested samples against: *S. aureus* (**a**), *E coli* (**b**), *E faecalis* (**c**), *K. Pneumoniae* (**d**), *S*. *typhimurium* (**e**), *C. Albicans* (**f**), *C. Glabrata* (**g**), and *A. brasiliensis* (**h**).

**Table 1 pharmaceutics-13-00477-t001:** The registered frequencies in the PEBSA, PEBSA_Ty, and thymol FTIR spectra and their absorption assignments.

FTIR Spectra					
PEBSA		PEBSA_Ty		Thymol	
Registered Frequency, cm^−1^	Absorption Assignment	Registered Frequency, cm^−1^	Absorption Assignment	Registered Frequency, cm^−1^	Absorption Assignment
2926.33	C-H stretching	3397.74	O-H stretching intermolecular bonded	3397.74	O-H stretching
2855.14	C-H stretching	2926.33	C-H stretching	2926.33	C-H stretching
2652.01	C-H stretching	2855.14	C-H stretching	1623.73	C=C stretching
1698.44	C=O stretching	1698.44 1623.73	C=O stretching C=C stretching	1520.41	C=C stretching
1465.50	C-H bending	1520.41	C=C stretching	1450	O-H bending
1240.99	C-O stretching	1465.50	C-H bending	1240.99	C-O stretching
1231.46	C-O stretching	1240.99	C-O stretching	1231.46	C-O stretching
1183.31	C-O stretching	1231.46	C-O stretching	1183.31	C-O stretching
919.18	C=C bending	1183.31	C-O stretching	1087	C-O stretching
681.97	C=C bending	1087.00	C-O stretching	919.18	C=C bending
531.26	C=C bending	919.18	C=C bending	807.14	C=C bending
		681.97	C=C bending	584.82	C=C bending
		531.26	C=C bending		

**Table 2 pharmaceutics-13-00477-t002:** Antimicrobial activity of the tested compounds against the references strains (mm).

Strains	Inhibition Zone (mm)		
	PEBSA	THYMOL	PEBSA_Ty
*Staphylococcus aureus ATCC25923*	-	7.793 ± 0.214	8.405 ± 0.282
*Escherichia coli ATCC25922*	-	10.527 ± 0.088	11.048 ± 0.546
*Enterococcus faecalis ATCC 29212*	-	7.550 ± 0.209	10.119 ± 0.151
*Klebsiella pneumonie ATCC 10031*	-	18.562 ± 0.085	19.837 ± 0.271
*Salmonella typhimurium ATCC 14028*	-	8.232 ± 0.008	7.135 ± 0.371
*Candida albicans ATCC10231*	-	10.091 ± 0.228	9.397 ± 0.267
*Candida glabrata ATCC 2001*	-	-	-
*Aspergillus brasiliensis ATCC9642*	-	18.288 ± 0.253	20.322 ± 0.501

**Table 3 pharmaceutics-13-00477-t003:** Antimicrobial activities of the investigated samples.

Strains	PEBSA		THYMOL		PEBSA_Ty	
	MIC (g/mL)	MBC (g/mL)	MIC (g/mL)	MBC (g/mL)	MIC (g/mL)	MBC (g/mL)
*S. aureus*	-^a^	-^a^	0.008	-^b^	0.008	-^b^
*E. coli*	-^a^	-^a^	0.008	-^b^	0.008	-^b^
*E. faecalis*	-^a^	-^a^	0.008	-^b^	0.008	-^b^
*K. pneumonie*	-^a^	-^a^	0.008	-^b^	0.008	-^b^
*S. Typhimurium*	-^a^	-^a^	0.008	-^b^	0.008	-^b^
*C. albicans*	-^a^	-^a^	0.008	-^b^	0.008	-^b^
*C. glabrata*	-^a^	-^a^	-^b^	-^b^	-^b^	-^b^
*A. brasiliensis*	-^a^	-^a^	0.008	-^b^	0.008	-^b^

**MIC**: minimum inhibitory concentration; **MBC**: minimum bactericidal concentration; -^a^ not tested; -^b^ not determined.

## Data Availability

All data supporting reported results are included in the article.

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
