# Peer review of "Synthesis of Poly(Ethylene Brassylate-Co-squaric Acid) as Potential Essential Oil Carrier"

_pharmaceutics, 2021, doi:10.3390/pharmaceutics13040477_

Round 1
Reviewer 1 Report
The study is devoted to a novel biocidal complex: thymol encapsulated by poly(ethylene brassylate-co-squaric acid). The idea of work and the main purposes of the study were clearly presented. All of this seems to moderately novel, the quality of presentation is acceptable, so I suggest publishing the manuscript, after some corrections, listed below.
Figure 1 has no reference in the text. Similarly, Fig. 3. What is more, the authors use "Figure" sometimes, another time abbreviation Fig. Make it, please, homogenous in the text.
Line 167: the equation should be labeled, similarly as in lines 179-179.
Line 176: the fonts are not uniform
From line 266: FTIR analysis is too shallow: the spectra should be described in more details (additionally to the table)
Figure 5b: the background should be subtracted from the spectrum – it is clearly seen, that the broad signal up to 2700cm-1 is a result of luminance.
Line 381: X-ray spectrum is visible at fig. 12, not 11
Author Response
Answers to the Reviewers
The manuscript, entitled:
Synthesis of poly(ethylene brassylate-co-squaric acid) as potential essential oil carrier
authors:
Aurica P. Chiriac, Alina Gabriela Rusu, Loredana Elena Nita, Ana-Maria Macsim, Nita Tudorachi, Irina Rosca, Iuliana Stoica, Daniel Tampu, Magdalena Aflori, Florica Doroftei
Firstly, we thank the Reviewers by helping us to improve our paper. The corrections were made and the manuscript was re-written accordingly with indications.
Comments from reviewer:
Figure 1 has no reference in the text. Similarly, Fig. 3. What is more, the authors use "Figure" sometimes, another time abbreviation Fig. Make it, please, homogenous in the text.
Line 167: the equation should be labeled, similarly as in lines 179-179.
Line 176: the fonts are not uniform
From line 266: FTIR analysis is too shallow: the spectra should be described in more details (additionally to the table)
Figure 5b: the background should be subtracted from the spectrum – it is clearly seen, that the broad signal up to 2700cm-1 is a result of luminance.
Line 381: X-ray spectrum is visible at fig. 12, not 11
Response to the reviewer’s comments (1)
- Figure 1 has no reference in the text. Similarly, Fig. 3. What is more, the authors use "Figure" sometimes, another time abbreviation Fig. Make it, please, homogenous in the text.
The corrections were made
- Line 167: the equation should be labeled, similarly as in lines 179-179.
The corrections were made
- Line 176: the fonts are not uniform
The corrections were made
- From line 266: FTIR analysis is too shallow: the spectra should be described in more details (additionally to the table)
FTIR figure was improved; also the following sentences were inserted at PEBSA characterization:
PEBSA spectrum present peaks corresponding to EB and SA comonomers, but also changes that occur as a result of the opening of the macrolactone cycle, are also registered. Thus, the specific peaks of OH from the squaric acid are no longer found in the copolymer spectrum, which attests to the formation of PEBSA. Also, asymmetric and symmetric CH2 bonds and carbonyl bonds (C=O), are found at 2926.33, 2855.14 and 1698.44 cm-1 respectively. Asymmetric and symmetric CH2 bonds and carbonyl bonds (C=O), are found at 2926.33, 2855.14 and 1698.44 cm-1 respectively; C-O and C-C bands at 1240.99 and 1231.46 cm-1, while asymmetric symmetric C-O-C bands are present at 1183.31 cm-1. Also, C=C bending from squaric acid are found at 919.18, 681.97, and 531.26 cm-1 respectively.
The following sentence referring to FTIR spectra of the bioactive complex is presented as well at PEBSA_Thymol bioactive complex characterization:
Thus, the supplementary peaks appeared in the FTIR spectra of PEBSA_Ty (Fig. 4 and Table 1) positioned at 3397.74 (O-H stretching intermolecular bonds), 1623.73 (C=C stretching), 1520.41 (C=C stretching), and 1087.00 (C-O stretching) cm-1, are attributed to the thymol compound and attest the formation of the bioactive complex.
-Figure 5b: the background should be subtracted from the spectrum – it is clearly seen, that the broad signal up to 2700cm-1 is a result of luminance.
The figure was improved and the background was subtracted and the cosmic ray removed from all spectra.
- Line 381: X-ray spectrum is visible at fig. 12, not 11
The correction was made

Reviewer 2 Report
The presented article describes the synthesis of a copolymer used as as efficient drug carrier vehicle of thymol. In the article some chemical and functional properties of obtained materials were done and the results were presented. Whole is written in good English style. Unfortunately, a lot of scientific deficiencies and omitted topics for consideration was observed in this work. Additionally, the scientific considerations are very sketchy. However, due to an interesting topic, the Reviewer gives the authors the opportunity to improve the article in order to increase its scientific value. Taking that into account I recommend major revision of this article. Below you can find detailed comments. Without their improvement, the article will have to be rejected in order not to underestimate the scientific rank of the journal. Introduction 1. The whole introduction is very badly done. The properties of the individual substrates are mainly described. Paragraphs are poorly connected thematically. There are no references to the current state of knowledge in the field of similar materials. How does the work of the authors differ from the current state of knowledge? What will be the final use of the received samples? The introduction should focus on the final application of the obtained samples. 2. Please, revise all the acronyms and provide the correct full name for them before using it. 3. The microbial strain name must be presented in italic. Materials and methods. 4. Please, improve Fig. 1. The resolution is not good. Also, use appropriate software to improve the chemical structures. 5. Can the drying process remove the residual organic solvent used in the synthesis? 6. The authors could use the DLS to determine the molar mass as well. 7. How was the thymol efficiency encapsulation determined? Provide more details in the experimental section. 8. Fig 2 needs to be improved like Fig. 1. 9. Is the first time that this copolymer prepared? The authors must compare their results with other papers. 10. FTIR - The analysis of the FTIR results is very chaotic and there are practically no scientific conclusions. Attention should be paid to the positions of individual peaks and their intensities in pure substrates (PEBSA and thymol) and mixture (PEBSA/thymol). The most important are reactive functional influenza. The FTIR figure must be improved as well. I can not see the numbers in the x-axis. 11. The authors need to compare their results. For example, you should associate the results obtained by different techniques of characterization. What is the relation between FTIR, NMR, and Raman results for example? 12. Fig. 9b must be improved as well. 13. Why there is no significant different in the DLS peaks around 500 nm? The peak ascribed to the polymer size, should not occur above 500 nm. 14. Please, check some typographical errors (bad usage symbols), non-uniform (different) notations. Please prevent unwanted word wrapping between number and unit. Please check whole manuscript and correct all wrong records (many times). 15. Fig. 15 must be improved. The results must be associated between them (all sections). The results must be compared with other results already reported in the literature. 16. The authors need to measure the encapsulation efficiency and carry out release studies of thymol to confirm the manuscript proposal. Why is better use the system polymer/thymol than pure thymol? 17. Conclusions Needs to be rewritten based on new results and manuscript novelty.Author Response
Answers to the Reviewers
The manuscript, entitled:
Synthesis of poly(ethylene brassylate-co-squaric acid) as potential essential oil carrier
authors:
Aurica P. Chiriac, Alina Gabriela Rusu, Loredana Elena Nita, Ana-Maria Macsim, Nita Tudorachi, Irina Rosca, Iuliana Stoica, Daniel Tampu, Magdalena Aflori, Florica Doroftei
Firstly, we thank the Reviewers by helping us to improve our paper. The corrections were made and the manuscript was re-written accordingly with indications.
Comments from reviewer:
The presented article describes the synthesis of a copolymer used as as efficient drug carrier vehicle of thymol. In the article some chemical and functional properties of obtained materials were done and the results were presented. Whole is written in good English style. Unfortunately, a lot of scientific deficiencies and omitted topics for consideration was observed in this work. Additionally, the scientific considerations are very sketchy. However, due to an interesting topic, the Reviewer gives the authors the opportunity to improve the article in order to increase its scientific value. Taking that into account I recommend major revision of this article. Below you can find detailed comments. Without their improvement, the article will have to be rejected in order not to underestimate the scientific rank of the journal. Introduction 1. The whole introduction is very badly done. The properties of the individual substrates are mainly described. Paragraphs are poorly connected thematically. There are no references to the current state of knowledge in the field of similar materials. How does the work of the authors differ from the current state of knowledge? What will be the final use of the received samples? The introduction should focus on the final application of the obtained samples. 2. Please, revise all the acronyms and provide the correct full name for them before using it. 3. The microbial strain name must be presented in italic. Materials and methods. 4. Please, improve Fig. 1. The resolution is not good. Also, use appropriate software to improve the chemical structures. 5. Can the drying process remove the residual organic solvent used in the synthesis? 6. The authors could use the DLS to determine the molar mass as well. 7. How was the thymol efficiency encapsulation determined? Provide more details in the experimental section. 8. Fig 2 needs to be improved like Fig. 1. 9. Is the first time that this copolymer prepared? The authors must compare their results with other papers. 10. FTIR - The analysis of the FTIR results is very chaotic and there are practically no scientific conclusions. Attention should be paid to the positions of individual peaks and their intensities in pure substrates (PEBSA and thymol) and mixture (PEBSA/thymol). The most important are reactive functional influenza. The FTIR figure must be improved as well. I can not see the numbers in the x-axis. 11. The authors need to compare their results. For example, you should associate the results obtained by different techniques of characterization. What is the relation between FTIR, NMR, and Raman results for example? 12. Fig. 9b must be improved as well. 13. Why there is no significant different in the DLS peaks around 500 nm? The peak ascribed to the polymer size, should not occur above 500 nm. 14. Please, check some typographical errors (bad usage symbols), non-uniform (different) notations. Please prevent unwanted word wrapping between number and unit. Please check whole manuscript and correct all wrong records (many times). 15. Fig. 15 must be improved. The results must be associated between them (all sections). The results must be compared with other results already reported in the literature. 16. The authors need to measure the encapsulation efficiency and carry out release studies of thymol to confirm the manuscript proposal. Why is better use the system polymer/thymol than pure thymol? 17. Conclusions Needs to be rewritten based on new results and manuscript novelty.
Response to the reviewer’s comments (2)
Introduction
- The whole introduction is very badly done. The properties of the individual substrates are mainly described. Paragraphs are poorly connected thematically. There are no references to the current state of knowledge in the field of similar materials. How does the work of the authors differ from the current state of knowledge? What will be the final use of the received samples? The introduction should focus on the final application of the obtained samples.
The introduction was re-written; the paragraphs are now connected thematically, and new references concerning the current state of knowledge in the field were added; it was underlined the fact that the study presents the synthesis of a new copolymer, its physicochemical characterization, and preliminary investigations concerning the thymol encapsulation are made in order of the future use of the copolymer_thymol complex as an effective antibacterial system.
- Please, revise all the acronyms and provide the correct full name for them before using it.
The acronyms were verified and their correct full name was presented before using them.
- The microbial strain name must be presented in italic.
The corrections were made
Materials and methods.
- Please, improve Fig. 1. The resolution is not good. Also, use appropriate software to improve the chemical structures.
The corrections were made: Fig. 1 was improved and ChemDraw ultra 12 was used for the chemical structures representation.
- Can the drying process remove the residual organic solvent used in the synthesis?
It was an omission in the presentation; like in the case of the bioactive complex the copolymer was freeze-dried and the procedure it is now mentioned in the article.
- The authors could use the DLS to determine the molar mass as well.
Thank you for recommendation; we will use the DLS system in the future and compare the molar mass of the copolymer with that of the homopolymer.
- How was the thymol efficiency encapsulation determined? Provide more details in the experimental section.
The purpose of the study was to achieve a polymer network capable for thymol encapsulation and further use of the new complex as an effective antibacterial system. Further investigations concerning the bioactive complex are ongoing. The following sentence was inserted in Materials and methods:
The recorded encapsulation efficiency of about 75% was obtained by UV absorption spectrophotometry from the water resulting after washing the PEBSA_Ty complex, by determining the absorbance values recorded at λ = 275 nm, and using the calibration curve previously performed.
- Fig 2 needs to be improved like Fig. 1.
The corrections were made: Fig. 2 was improved and ChemDraw ultra 12 was used for the chemical structures representation.
- Is the first time that this copolymer prepared? The authors must compare their results with other papers.
Yes, this is a new polymer structure and as result the synthesized copolymer was strictly compared several times just with the homopolymer, poly(ethylene brassylate); in this context the following sentence was inserted in the Introduction part:
The novelty of this research is the synthesis of poly(ethylene brassylate-co-squaric acid) (PEBSA), a new biodegradable macromolecular compound with improved functionality, and the attempt of its encapsulation with thymol (a natural monoterpenoid phenol found in oil of thyme), a bioactive molecular compound with biocompatible and antioxidant character, strong antiseptic properties, with antimicrobial activities against Escherichia coli, Staphylococcus aureus and Streptococcus mutans.
- FTIR - The analysis of the FTIR results is very chaotic and there are practically no scientific conclusions. Attention should be paid to the positions of individual peaks and their intensities in pure substrates (PEBSA and thymol) and mixture (PEBSA/thymol). The most important are reactive functional influenza. The FTIR figure must be improved as well. I can not see the numbers in the x-axis.
FTIR figure was improved; also the following sentences were inserted at PEBSA characterization:
PEBSA spectrum present peaks corresponding to EB and SA comonomers, but also changes that occur as a result of the opening of the macrolactone cycle, are also registered. Thus, the specific peaks of OH from the squaric acid are no longer found in the copolymer spectrum, which attests to the formation of PEBSA. Also, asymmetric and symmetric CH2 bonds and carbonyl bonds (C=O), are found at 2926.33, 2855.14 and 1698.44 cm-1 respectively. Asymmetric and symmetric CH2 bonds and carbonyl bonds (C=O), are found at 2926.33, 2855.14 and 1698.44 cm-1 respectively; C-O and C-C bands at 1240.99 and 1231.46 cm-1, while asymmetric symmetric C-O-C bands are present at 1183.31 cm-1. Also, C=C bending from squaric acid are found at 919.18, 681.97, and 531.26 cm-1 respectively.
Also, the following sentence referring to FTIR spectra of the bioactive complex was included at PEBSA_Thymol bioactive complex characterization:
Thus, the supplementary peaks appeared in the FTIR spectra of PEBSA_Ty (Fig. 4 and Table 1) positioned at 3397.74 (O-H stretching intermolecular bonds), 1623.73 (C=C stretching), 1520.41 (C=C stretching), and 1087.00 (C-O stretching) cm-1, are attributed to the thymol presence, and attest the formation of the bioactive complex.
- The authors need to compare their results. For example, you should associate the results obtained by different techniques of characterization. What is the relation between FTIR, NMR, and Raman results for example?
The following sentence was inserted in the manuscript along with the other requested additions:
All the performed spectroscopic analyses confirm the synthesis of PEBSA copolymer, from the breaking cycle of EB evidenced by NMR spectra, to correlated and complementary peaks from FTIR and Raman spectra, which attest the asymmetric and symmetric CH2 and carbonyl bonds presence from chains, as well the intra- and intermolecular physical interactions within the macromolecular chains. These results are sustained by other investigation too. [24]
- Fig. 9b must be improved as well.
Fig. 9b was improved
- Why there is no significant different in the DLS peaks around 500 nm? The peak ascribed to the polymer size, should not occur above 500 nm.
You have right: from the figure below the peak ascribed to the polymer size did not appear above 500 nm and this behavior registered after 24 hours from the complexation between PEBSA and Thymol; this behavior attests the compatibility between the two structures, as well as the complex formation. We can consider initially in the case of the copolymer a supramolecular structure provided by intra- and intermolecular physical bonds; subsequently due to the affinity of the polymer towards thymol, new physical connections are made, which ensures the encapsulation and packing of thymol in the polymer matrix concretized in the diminution of the particle size.
Size distribution of PEBSA copolymer, PEBSA_Ty complex and Thymol
- Please, check some typographical errors (bad usage symbols), non-uniform (different) notations. Please prevent unwanted word wrapping between number and unit. Please check whole manuscript and correct all wrong records (many times).
The manuscript was revised and verified for typographical errors, non-uniform notations, and they were corrected.
- Fig. 15 must be improved. The results must be associated between them (all sections). The results must be compared with other results already reported in the literature.
Fig. 9b was improved
- The authors need to measure the encapsulation efficiency and carry out release studies of thymol to confirm the manuscript proposal. Why is better use the system polymer/thymol than pure thymol?
The following sentence was inserted in Materials and methods:
The recorded encapsulation efficiency of about 75% was obtained by UV absorption spectrophotometry from the water resulting after washing the PEBSA_Ty complex, by determining the absorbance values recorded at λ = 275 nm, and using the calibration curve previously performed.
The purpose of the study was to confirm the possibility of thymol encapsulation in the network of this new poly(ethylene brassylate-co-squaric acid) structure in order to obtain a new complex as an effective antibacterial system. Further investigations concerning this new complex and its applicability are underway.
- Conclusions Needs to be rewritten based on new results and manuscript novelty.
The conclusions were rewritten and the new results and the novelty were mentioned including the increase of the antimicrobial activity in case of the PEBSA_Ty complex use against some microbial strains

Round 2
Reviewer 2 Report
The authors have significantly improved the manuscript. However, they did not respond one question.
Why is better to use the polymer/thymol system than the pure thymol?
The authors can respond this question by evaluating the release assay of thymol from the polymer material.
Author Response
Answers to the Reviewers
The manuscript, entitled:
Synthesis of poly(ethylene brassylate-co-squaric acid) as potential essential oil carrier
authors:
Aurica P. Chiriac, Alina Gabriela Rusu, Loredana Elena Nita, Ana-Maria Macsim, Nita Tudorachi, Irina Rosca, Iuliana Stoica, Daniel Tampu, Magdalena Aflori, Florica Doroftei
Thank you again by helping us to improve our paper.
Concerning your question:
Why is better to use the polymer/thymol system than the pure thymol?
Answer:
As we mentioned before:
The purpose of the study was to confirm the possibility of thymol encapsulation in the network of this new poly(ethylene brassylate-co-squaric acid) structure in order to obtain a new complex as an effective antibacterial system, and we intend to use this system in a gel patch device. Further investigations concerning this new complex and its applicability are underway and the new data constitute the results of another report.
Also, from the determined antimicrobial activities of the investigated samples (from Table 2 of the article) it is clearly observed that increased antimicrobial activities correspond to PEBSA_Ty complex when it was tested on the majority of strains namely Staphylococcus aureus ATCC25923, Escherichia coli ATCC25922, Enterococcus faecalis ATCC 29212, Klebsiella pneumonie ATCC 10031, Aspergillus brasiliensis ATCC9642.
Thank you for your understanding.
Strains |
Inhibition zone (mm) |
||
PEBSA |
THYMOL |
PEBSA_Ty |
|
Staphylococcus aureus ATCC25923 |
- |
7.793 ± 0.214 |
8.405 ± 0.282 |
Escherichia coli ATCC25922 |
- |
10.527 ± 0.088 |
11.048 ± 0.546 |
Enterococcus faecalis ATCC 29212 |
- |
7.550 ± 0.209 |
10.119 ± 0.151 |
Klebsiella pneumonie ATCC 10031 |
- |
18.562 ± 0.085 |
19.837 ± 0.271 |
Salmonella typhimurium ATCC 14028 |
- |
8.232 ± 0.008 |
7.135 ± 0.371 |
Candida albicans ATCC10231 |
- |
10.091 ± 0.228 |
9.397 ± 0.267 |
Candida glabrata ATCC 2001 |
- |
- |
- |
Aspergillus brasiliensis ATCC9642 |
- |
18.288 ± 0.253 |
20.322 ± 0.501 |
